# The Clinical Efficacy of Adding Ceftazidime/Avibactam to Standard Therapy in Treating Infections Caused by Carbapenem-Resistant *Klebsiella pneumonia* with blaOXA-48-like Genes

**DOI:** 10.3390/antibiotics13030265

**Published:** 2024-03-16

**Authors:** Al Maamon R. Abu Jaber, Bilgen Basgut, Ali Abdullah Hawan, Ali Amer Al Shehri, Sultan Ahmad AlKahtani, Nehad J. Ahmed, Abdikarim Abdi

**Affiliations:** 1Department of Clinical Pharmacy, Faculty of Pharmacy, Near East University, Nicosia 99138, Northern Cyprus TR-10 Mersin, Turkey; abdikarim.abdi@yeditepe.edu.tr; 2Department of Pharmacology, Faculty of Pharmacy, Baskent University, Ankara 06790, Turkey; bilgenbasgut@gmail.com; 3The Armed Forces Hospitals Southern Region AFHSR, Khamis Mushait 62413, Saudi Arabia; binhawan@gmail.com (A.A.H.); aaas1400@hotmail.com (A.A.A.S.); s_kahtani@hotmail.com (S.A.A.); 4Department of Clinical Pharmacy, College of Pharmacy, Prince Sattam Bin Abdulaziz University, Alkharj 11942, Saudi Arabia; n.ahmed@psau.edu.sa; 5Department of Clinical Pharmacy, Faculty of Pharmacy, Yeditepe University, İstanbul 34755, Turkey

**Keywords:** carbapenem resistance, *Klebsiella pneumonia*, OXA-48-like genes, ceftazidime/avibactam, clinical efficiency

## Abstract

Ceftazidime/avibactam (CAZ-AVI) is FDA-approved for managing infections caused by resistant gram-negative bacilli, particularly infections via carbapenem-resistant *Enterobacterales* pathogens. The clinical data are still limited, particularly those in Saudi Arabia. The present study is a retrospective cohort study that was carried out at the Armed Forces Hospital in the southern region of Saudi Arabia to compare the clinical and microbiological outcomes for CAZ-AVI-treated patients as monotherapy and as an add-on to standard therapy for carbapenem-resistant *Klebsiella pneumonia* (CRKP) OXA-48 infections to those treated with standard drugs. The study included CRKP OXA-48-like infected patients who were administered antibiotics for more than seven days from 1 August 2018 to May 2023. Patients’ baseline characteristics and demography were extracted from the clinical records, and their clinical/microbiology efficiencies were assessed as per the corresponding definitions. Univariate and multivariate logistic regressions were conducted to identify the potential independent variable for CAZ-AVI efficiency. A total of 114 patient files were included for the evaluation. Among these patients, 64 used CAZ-AVI combined with standard therapy and were included in the intervention group, and 50 of them used standard therapy and were included in the comparative group. Following analysis, CAZ-AVI’s clinical success was 42.2% (*p* = 0.028), while the intervention versus comparative groups showed decreased 30-day all-cause mortality (50.0% versus 70.0%; *p* = 0.036) and infection recurrence (7.8% versus 24.0%; *p* = 0.019), as well as substantially increased rates of microbial eradication (68.8% versus 42.0%; *p* = 0.007). CAZ-AVI add-on therapy rather than monotherapy showed statistically significant favored clinical and microbial outcomes over the standard therapy. Furthermore, sex (female %), ICU admission, and fever were negatively associated with patients’ 30-day all-cause mortality, serving as independent negative factors. Only fever, CRP bio levels, inotropes, and ICU admissions were significant predictors influencing the CAZ-AVI’s clinical efficiency. The duration of CAZ-AVI therapy positively influenced CAZ-AVI’s microbial eradication, while both WBC counts and fever experiences were negative predictors. This study shows the effective usage of CAZ-AVI against CRKP OXA-48-like infections. The influencing independent variables depicted here should recommend that clinicians individualize the CAZ-AVI dose based on co-existing risk factors to achieve optimal survival and efficacy. Prospective multicenter and randomized control studies are recommended, with individualized CAZ-AVI precision administration implemented based on patients’ characteristics.

## 1. Introduction

The gram-negative bacterium *Enterobacterales* can cause a variety of healthcare-associated illnesses, including meningitis, bloodstream infections, wound or surgical site infections, and pneumonia. There has been a rise in the number of cases of bacteremia caused by *Enterobacterales* all over the world, particularly those caused by resistant strains of *Klebsiella pneumonia* [1]. Antibiotic resistance in *K. pneumonia* bacteria is on the rise, most recently within the carbapenem antibiotic class [2]. Unfortunately, when it comes to gram-negative infections that are resistant to other antibiotics, carbapenem medicines are frequently the final resort [2]. Since *Enterobacterales* produce extended-spectrum beta-lactamase (ESBL), which is a common cause of infections, antibiotics like carbapenem are frequently used to treat these infections [3]. However, overuse or improper handling of these antibiotics has led to the emergence of isolates that are resistant to carbapenem.

Antibiotic resistance is a serious threat to human health and an increasing concern. By the year 2050, it is anticipated that antibiotic-resistant diseases will cause 10 million annual deaths worldwide [4]. The management of infections caused by carbapenem-resistant *K. pneumonia* (CRKP) is challenging owing to high antibiotic resistance levels, limited available therapeutic options, and inconsonant optimal treatment duration [5,6]. The World Health Organization lists the bacterium *K. pneumonia* as a critical priority antibiotic-resistant bacteria pathogen, being a substantial contributor to both hospital- and community-acquired illnesses [7]. The emergence of carbapenem resistance appears to be a global phenomenon and occurs in clinical, urban, and agricultural settings [8,9,10,11,12,13]. The mortality of patients infected with CRKP is three times higher than that of patients infected with susceptible *K. pneumonia* strains, ranging from 30 to 44% and startlingly approaching 70% in the case of bacteremia [14,15,16,17,18]. According to the 2019 CDC antibiotic resistance report [19] and the China Antimicrobial Surveillance Network [20], most carbapenem-resistant *Enterobacterales* are CRKP, with lower respiratory infections being the most frequent cases. Notably, CRKP is associated with a greater death rate, with a roughly 2-fold increase in mortality when compared to carbapenem-susceptible enterobacterial infections [19,21].

It has been widely reported that the fundamental mechanism of carbapenem resistance within CRKP is through the production of carbapenemases enzymes [22]. Clinically significant carbapenemases within *Enterobacterales* are categorized into serine-β-lactamases with Ambler sub-classes (A and D) enzymes or metallo-β-lactamases with Ambler sub-classes (B) [23]. Genetically encoded, class A carbapenemases have originated from several chromosomal genes (*BIC-1*, *FPH-1*, *NmcA*, *PenA*, *SFC-1*, *SHV-38*, and *SME*) or plasmids (*FRI*-1, *GES*, and *KPC*), where KPC strains have been highly associated with CRKP [24]. On the other hand, class D carbapenemases demonstrated wide distribution among *Enterobacterales*, including *K. pneumonia,* being encoded by *OXA-48* and eleven identified variants (*OXA-48-like*), showing relevant geographical variations [25,26]. Since the discovery of *OXA-48* carbapenemase in Turkey in 2004, these OXA-48 strains have been frequently documented within nosocomial outbreaks throughout the world, particularly in the Mediterranean region [27]. 

Generally, penicillin, cephalosporins, and carbapenems that are currently on the market show no in vitro activity against this resistant pathogen and CRKP bloodstream infections [28]. Therapeutic combinations between third-generation cephalosporin “ceftazidime” and non-β-lactam/lactamase suicidal inhibitor “avibactam” (CAZ-AVI) have demonstrated microbiological and clinical efficiencies against class A, class C, and a few class D (*OXA-48*) *K. pneumonia* carbapenemases, but not class B metallo-β-lactamases [29]. Treatment with CAZ-AVI has been approved by US-FDA, EMA, and Chinese-FDA for managing complicated urinary tract (including pyelonephritis), intra-abdominal, and hospital-acquired pneumonia infections [5]. There is accumulating evidence that CAZ-AVI can be utilized to treat infections caused by resistant gram-negative bacilli, particularly carbapenem-resistant *Enterobacterales* infections [30,31,32,33,34]. Systematic reviews furnished clinical evidence for the beneficial use of CAZ-AVI for hospitalized patients with multi-drug-resistant *K. pneumonia* and carbapenem-resistant *Enterobacterales* [35,36]. Real-world investigations showed that this novel β-lactam/lactamase inhibitor combination is the best therapeutic option for CRKP as it reduced 30-day mortality in bacteremia, carbapenem-resistant *Enterobacterales*-associated clinical failure, and 14-day microbiological failure rates [34,37,38]. Recently, sole or combined treatment of CAZ-AVI with aztreonam demonstrated good antimicrobial and synergistic bacteriostatic/bactericidal activities against KPC-, IMP-, OXA-, and/or NDMI-producing strains [39].

Despite the encouraging efficiency of CAZ-AVI within carbapenem-resistant *Enterobacterales* and CRKP infections, clinical experience is still lacking, and further real-world investigations are needed. Carbapenemases are abundant in *K. pneumonia* isolates in Saudi Arabia, where reports highlighted that the most common carbapenemases are OXA-48, followed by the New Delhi metallo-lactamase [40,41]. Studies comparing the outcomes of patients in Saudi Arabia with carbapenem-resistant *Enterobacterales* (CRE) infections treated with CAZ-AVI versus other regimens are still lacking. In these regards, our study aimed to retrospectively investigate the CAZ-AVI’s microbiological and clinical efficiency as well as the mortality of CRKP OXA-48-like infected patients by assessing their clinical results upon treatment with CAZ-AVI as monotherapy and as an add-on to standard therapy as compared to those receiving other drugs.

## 2. Results

### 2.1. Patients’ Characteristics

Within the time period between 1 August 2018 and 1 May 2023, a total of 228 CRKP patients were admitted to the Armed Forces Hospitals, in the Saudi southern region, Khamis Mushait, Saudi Arabia. Based on the adopted inclusion and exclusion criteria, a final total number of 114 patients with *K. pneumonia* OXA-48-like genes were included in the study for evaluation. Patients who were below 18 years old, who died prior to antibiotic initiation, or who used antibiotic regimens for less than 5 days were excluded. Among these patients, 64 were given CAZ-AVI alone or in combination with the standard therapy (intervention group), while 50 of them were administered the standard therapy and were included as the comparative control group. The demographic features and baseline clinical characteristics of the CRKP OXA-48-like infected patients are provided in Table 1. The median patient age was 71 (minimum 20.0–maximum 102.0) years old, with the majority of the patients sex being male (66; 57.9%). Out of the included patients, a total of 85 (74.6%) were admitted to the ICU for medical condition management of which 42 (65.6%) and 43 (86.0%) of the intervention and control groups were ICU-admitted. Nearly 49% of the patients have received inotropes as vasoactive agents, including dopamine, epinephrine, norepinephrine, and/or vasopressin.

The proportions for sites of infection were 30 (26.3%) for infections within the bloodstream, 31 (27.2%) respiratory tract, 21 (18.4%) soft tissues, and the lowest proportions were 4 (3.5%) for the urinary tract. Almost 25% of the included patients were presented with multiple-site infections with K. pneumonia OXA-48-like strains. Notably, the control patient group was presented with lower proportions for soft tissue infections compared to those of intervention one: 3 (6.0%) versus 18 (28.1%). Patients were presented with several types of co-morbidities, where higher patient proportions were assigned with 83 (72.8%) for cardiovascular disease, 69 (60.5%) for diabetes mellitus, 37 (32.5%) for respiratory disease, and 46 (40.4%) for renal diseases. No significant differences were depicted between the intervention and control (*p* > 0.05) in terms of co-morbidity proportions. Patients’ baseline lab tests and clinical signs were measured in terms of fervescence (≥38.0 °C; 100.4 °F for 48 h), white blood cell (WBC) counts, neutrophil counts, and C-reactive protein (CRP) blood levels. Among the included patients, average WBCs and neutrophil counts were 12.0 × 10^9^/L (±0.7) and 12.7 × 10^9^/L (±2.0), respectively. Higher blood levels of CRP were observed with the control compared to the intervention, 169.3 mg/L (±21.4) versus 80.1 mg/L (±8.7), respectively. Fever was accounted for within 54.4% of the total analyzed patients, where higher proportions were also accounted for in the controls, 35 (70.0%).

### 2.2. Antibiotic Medications 

At the final analysis, CRKP OXA-48-like infection patients received antibiotic regimens over an average time period of 14.0 (±0.7) days, starting from the first-positive cultures for the OXA-48-like gene (Figure 1A). Antibiotics received were either monotherapy for 24 (21.1%) of all patient cases or even a combination of two agents or more drugs within 49 (43.0%) and 41 (36.0%) of the total analyzed patients, respectively (Figure 1B). The standard antibiotic regimens included aminoglycosides for 9 (7.9%), aztreonam for 4 (3.5%), colistin for 61 (53.5%), and tigecycline for 45 (39.5%) of the total patient cases. All standard antibiotic agents were significantly indifferent between the intervention and control groups, except for meropenem being more frequently administered (48; 96.0%), almost to all cases in the comparative patient group (Figure 1C). Within the intervention group, the CAZ-AVI drug was given as definitive therapy rather than empirically (starting from first-positive cultures for the OXA-48-like gene) being administered alone as monotherapy or in combination with one additional drug or even multiple drugs for 13 (21.9%), 28 (43.8%), and 23 (35.9%) of the intervention patient cases, respectively (Figure 1D). The proportions of the added antibiotic type to CAZ-AVI are as follows: aminoglycosides for 6 (9.4%), aztreonam for 3 (4.7%), colistin for 32 (50.0%), meropenem for 8 (12.5%), and tigecycline for 23 (35.9%) of the intervention group cases (Figure 1E). Thus, higher frequencies were depicted for the combination of colistin and tigecycline as standard antibiotic therapy in combination with CAZ-AVI.

Doses of administered CAZ-AVI ranged from 940 mg to 2500 mg, being provided once daily (q24h), every other day (EOD), twice per day (q12h), or even three times a day (q8h). CAZ-AVI dosing was found to be in concordance with the clinical guidelines and practices relying on the patients’ physiological status, whereas the usual dosing of 2500 mg q8h was for admitted patients with creatinine clearance (CrCl) above 50 mL/min. Patients with reported CrCl > 30-to-50 mL/min were provided with 1250 mg (q8h). Patients with reported CrCl > 15-to-30 mL/min received 940 mg q12h, while CAZ-AVI was reported with the same dose yet every 24 h (q24h) at CrCl > 5–15 mL/min. A dose of 940 mg EOD was reported in patients with CrCl ≤ 5 mL/min, whereas critically ill patients with acute kidney injuries received 1250 mg q12h CAZ-AVI on prolonged intermittent renal replacement therapy (PIRRT) days. Notably, the median first CAZ-AVI dosage across the treated patients was 1.25 g, while the cumulative dosage over the antibiotic course was 39.74 g with an average of 3.75 g/day (Figure 1F). As per patients’ physiological status, the most observed CAZ-AVI dose regimen was 2500 mg q8h, followed by 940 mg q12h, whereas the 1250 mg q12h dose regimen was the least frequently received with 27 (42.19%), 17 (26.56%), and 2 (3.13%) patient cases, respectively (Figure 1G). 

### 2.3. Parameters Associated with CAZ-AVI Clinical and Microbiological Outcomes

Following the study analysis, 30-day all-cause mortality was observed in 51.6% of the patients in the intervention group, where 32 patients died. Comparatively, 66% of the patients within the control group died from any cause of death that happened within the 30 days of bacterial isolates [42]. Based on the adopted definitions of patients’ secondary outcomes, 27 (42.0%) patients of the 64 intervention patients had clinical success with fever remission as well as WBCs, procalcitonin, and CRP blood level normalization [43,44]. On the other hand, 11 patients over 50 (22.0%) of the comparative group had the same defined clinical success. Regarding microbial eradication, 44 CAV-AVI patients (68.0%) were eradicated from the target microorganism, showing two consecutive negative cultures from the same and different sites [45], whereas 21 control patients (42.0%) had microbial eradication. Only 5 (7.8%) patients among the CAV-AVI administered group and 12 (24%) patients among the comparative group had bacterial recurrence, having bacteremia with the same species and susceptibility pattern as the index blood isolate, following at least one negative microbe growth [44]. Notably, the presented study showed statistically significant differences for patients’ clinical and microbiological outcomes between the intervention and comparative groups at *p*-values < 0.05 (Table 2).

Dissecting the data as per antibiotic regimens (monotherapy and add-on therapy) has furnished interesting findings. Under monotherapy conditions, the defined clinical outcomes were at higher case percentages for the CAZ-AVI-treated OXA-48-like CRKP patients compared to the standard therapy in terms of clinical success (46.2% versus 27.3%) and 30-day all-cause mortality (69.2% versus 54.5%). Intervention CAZ-AVI monotherapy further depicted microbiological outcomes at higher case percentages for microbial eradication (68.8% versus 42.0%) and lower ones for bacterial recurrences (7.8% versus 24.0%) compared to the corresponding standard therapy. Nonetheless, a statistical analysis of the depicted differential outcomes highlighted no significant differences (*p* > 0.05) between the intervention and control groups under monotherapy. On the contrary, statistically significant differences were depicted across both the clinical and microbial outcomes within OXA-48-like CRKP cases receiving CAZ-AVI as an add-on to standard therapy. Lower 30-day all-cause mortality rates were seen with CAZ-AVI add-on therapy (45.1%) in relation to the controls (74.4%; *p* = 0.009) as only 23 patients died for the earlier therapy. Clinical success was statistically significant at higher rates for the CAZ-AVI add-on group, reaching 41.2% (*p* = 0.043), where 21 versus only 8 patients depicted fever remission and WBC/C-reactive protein normalization. Microbial outcomes, in terms of microbial eradication, were nearly 1.5-fold higher in patients who received CAZ-AVI add-on therapy compared to those who received only the standard therapy (66.7% versus 43.6%; *p* = 0.034). Most interestingly, bacterial recurrences were at much higher rates (almost 7-folds) for the control patients (20.5%) in relation to those who received the CAZ-AVI add-on therapy (3.9%) at *p* = 0.018. 

#### 2.3.1. Clinical Efficiency in CAZ-AVI Patient Group

Comparing patients’ characteristics, demography, and CAZ-AVI usage between the clinically successful and failure patients of the intervention group has revealed significant differences. Through the univariate analysis, CAZ-AVI successfully treated patients were of a lower female %, with fewer ICU admissions, fewer inotrope administrations, lower average WBC—neutrophil—CRP counts, and fewer fever presentations than patients of the treatment failure group (*p* < 0.05). On the contrary, the same treatment-successful patients had significantly higher frequencies of soft tissue infections and CAZ-AVI usage in terms of higher cumulative dosages and longer antibiotic duration compared to the therapy failure group (Table 3). The findings for the deceased versus survived CAZ-AVI patients are quite comparable to the above-described clinically successful/failure groups. Concerning female sex, ICU stays, inotrope administrations, sepsis comorbidity, and WBC—neutrophil—CRP counts, values were statistically lower for the survivors versus those of the deceased. Nonetheless, survival patients expressed higher incidences of soft tissue infection sites and cumulative CAZ-AVI dosages in relation to the deceased patients (Table 3). 

#### 2.3.2. Microbiological Efficiency in CAZ-AVI Patient Group

A univariate analysis was also applied for patients’ characteristics, demography, and CAZ-AVI usage between patients who expressed a clearance of the CRKP OXA-48-like strain and those who did not (microbial eradication versus persistent CAZ-UVI patients). Patients who still had positive CRKP OX-48-like strain microbial cultures were of higher female %, inotropes receival, WBC—neutrophil—CRP counts, fever experiences, and COVID-19 comorbidity. Contrarily, higher incidences of soft tissue infection sites as well as admissions to the ICU for treatment were significantly related to CRKP OXA-48-like clearance following a CAZ-AVI treatment course (Table 4). These microbial-eradicated patients further received more inotropes and higher CAZ-AVI cumulative dosages compared to the microbial-persistent group. Finally, bacterial relapse within CAZ-AVI patients was associated with only significantly lower WBCs and CRP blood counts than those of the bacteremia-receded patients (Table 4).

Further analysis using multivariate logistic regression proceeded to identify the independent factors and covariates (predictors) associated with risk or protective factors for the clinical and microbiological outcomes of CAZ-AVI-based therapy. Analysis findings in Table 5 showed that sex (female %), ICU admission, and fever were negatively associated with patients’ mortality, serving as independent negative factors for increasing patients’ survival with respective 0.105, 0.141, and 0.080 odds ratios (ORs). A multivariate logistic regression analysis further confirmed that fever experiences in patients, CRP levels, and inotropes were the significant independent negative factors (risk factors) that could impact the CAZ-AVI clinical efficiency at adjusted odds ratios of 0.004, 0.987, and 0.051, respectively. Only ICU admission was positively associated with clinical efficiency, serving as a positive predictor (OR = 21.183). Regarding microbial eradication, the duration of the CAZ-AVI therapy positively influenced the clearance of the CRKP OXA-48-like strain (OR = 1.446). On the contrary, both WBC counts and fever experiences were depicted as significant, independent covariates/factors that negatively impacted microbial eradication (ORs = 0.747 and 0.013, respectively). Interestingly, the impact of neither neutrophil counts nor CRP bio levels was depicted as significant, independent covariates for bacterial relapse, while the other factor was kept constant throughout the multivariate logistic regression analysis. Associations between significant independent factors/covariates and each clinical and microbiological outcome are illustrated via conditional estimation/prediction plots at their respective 95% confidence intervals within Figure 2.

To ensure the precision of the conducted multivariable logistic regression analysis, the risk of multicollinearity among the independent variables (predictors) was evaluated. Multicollinearity can compromise the statistical power of the regression model, causing estimation coefficients and *p*-values that are highly sensitive toward small model alterations [46,47]. Estimating the tolerance indices (TIs) and variance inflation factors (VIFs) for each independent predictor within the model was adopted to check the multicollinearity assumption. As a rule of thumb, VIFs of unity values (i.e., 1) and tolerance > 0.1 confer limited correlations among independent variables, while VIFs values below 5 and tolerance > 0.2 suggest moderate correlations, yet they warrant no corrective measures. However, VIFs above 10 and tolerance < 0.1 assume critical multicollinearity levels, with the assumptions being highly violated [46,48]. Throughout the adopted regression models, VIFs for independent variables were less than 3.5, with high tolerance indices at >0.6 in average (Table 5), conferring a good estimation of coefficients with *p*-values being unquestioned.

## 3. Discussion

Infections by CRKP, including OXA-48-like producing enzymes, as some of the most common carbapenem-resistant *Enterobacterales* impose great public health challenges owing to the limitedly available therapeutic regimens, increased prevalence, and high morbidity/mortality rates [49,50]. Additionally, patients’ mortality rates were reported to increase following the inappropriate management of such carbapenem-resistant infections [51]. The healthcare burden of OXA-48-like infections has also been highlighted by a three-year worldwide surveillance (2016–2018) where isolates with OXA-48-like pathogens have increased from 0.5% to 0.9%, with *K. pneumonia* being the primary microorganism in Europe and vicinal countries [52]. The surveillance study showed that the co-existence of other mutations/alterations like OmpK35, OmpK36, and/or blaCTX-M-15 within most OXA-48 isolates was associated with decreased susceptibility rates for meropenem and/or vaborbactam.

Before 2015, frontline therapies for carbapenem-resistant OXA-48-like CRKP included combinations of drugs with high rates of toxicity (aminoglycosides and colistin), poor pharmacokinetics (aminoglycosides, colistin, and tigecycline), and/or known microbiological resistance (carbapenem) [25]. Following the FDA approval of CAZ-AVI in 2015, this novel lactam/lactamase inhibitor provides significant improvement over earlier treatment regimens against *K. pneumonia* carbapenemases, where such a pathogen represents the primary source of carbapenem-resistant *Enterobacterales* infections in the United States [36]. Shirley et al. stated that CAZ-AVI has excellent in vitro activity against many important gram-negative pathogens, including many extended-spectrum β-lactamase-, AmpC-, *K. pneumonia* carbapenemase-, and OXA-48-producing *Enterobacterales* and drug-resistant *Pseudomonas aeruginosa* isolates [44]. Based on the INFORM global surveillance program over two time periods (2012–2014) and (2015–2017), it has been observed that CAZ-AVI demonstrated efficacy against a majority of carbapenem-resistant *Enterobacterales* isolates producing KPC and OXA-48-like enzymes [53,54]. The guidelines by both the Infectious Diseases Society of America [45,55] and the European Society of Clinical Microbiology and Infectious Diseases [56] stated the preferentiality of using CAZ-AVI against carbapenem-resistant *Enterobacterales*-producing OXA-48-like enzymes as the first treatment line showing in vitro activity against Ambler class A (KPC) and specific class D (OXA-48) carbapenemases, yet they were ineffective against MBL producers. The guidelines further highlight cefiderocol as an alternative option, with neither vaborbactam–meropenem nor cilastatin–imipenem–relebactam being effective against carbapenem-resistant *Enterobacterales*-producing OXA-48-like enzymes.

Despite the encouraging clinical data for CAZ-AVI in CRKP-infected patients, clinical experiences within Saudi Arabia are still limited. The present study assessed the microbiological and clinical efficiencies of CAZ-AVI add-on therapy to the standard antibiotic regimen in OXA-48-like CRKP-infected patients within a single-center cohort. Compared to standard therapy, the study’s primary findings highlighted the benefits of CAZ-AVI, including increased remission/clinical success (42.2% versus 22.0%; *p* = 0.028), decreased 30-day all-cause mortality (50.0% versus 70.0%; *p* = 0.036), decreased infection recurrence (7.8% versus 24.0%; *p* = 0.019), as well as substantially increasing the rate of microbial eradication (68.8% versus 42.0%; *p* = 0.007). Lower % ICU admission rates (65.6% versus 86.0%; *p* = 0.017) and higher soft tissue infections (28.1% versus 6.0%; *p* = 0.003) were also assigned to the intervention patients compared to the controls. Most interestingly, CAZ-AVI add-on therapy to standard antibiotic regimens rather than monotherapy showed statistically significant favored clinical and microbial outcomes over the control group. Furthermore, sex (female %), ICU admission, and fever were negatively associated with patients’ 30-day all-cause mortality, serving as independent negative factors. Fever, CRP bio levels, and inotropes were the significant risk factors influencing CAZ-AVI’s clinical efficiency, while only ICU admission was positively associated with clinical efficiency, serving as a positive predictor. The duration of CAZ-AVI therapy positively influenced the CAZ-AVI’s microbial eradication, while both WBC counts and fever experiences were negative, independent covariates/factors. Notably, our observed data are consistent with reported real-world research data as well as molecular mechanistic aspects of eradicating OXA-48-like CRKP strains.

Within worldwide single-center reports and/or small-sample-sized studies, the evidence supports CAZ-AVI’s excellent effectiveness in managing carbapenem-resistant *Enterobacterales* infections, including CRKP ones, which have been highlighted in a range from 33.3% to 81.8% [34,57,58,59]. Reported studies have further highlighted CAZ-AVI-associated microbiological clearances ranging from 36.7% to 79.5% among several kinds of carbapenem-resistant *Enterobacterales* infections [37,60,61]. Specific predictors such as chest infections and an INCREMENT-CPE score above seven points have also been highlighted to negatively impact the 14-day clinical success of CAZ-AVI treatment for KPC-producing *K. pneumonia* [62]. Our observations also depicted moderate clinical success/efficiency of nearly 42% and microbial eradication of almost 69% for the CAZ-AVI-treated patients, with several predictors influencing these findings. Domestic studies have illustrated comparable findings. A study by Alqahtani et al. reported that among adult CAZ-AVI-treated patients admitted to King Abdul-Aziz Medical City, Saudi Arabia, between 2018 and 2020, they showed an overall clinical cure rate of 78% versus 42.2% in the comparative group [63]. This study highlighted that most patients had hospital- or ventilator-acquired pneumonia, with *K. pneumonia* being the most common causative pathogen, and the majority of isolates contained OXA-48 enzymes (81%). Another study reported by Alraddadi et al. identified the OXA-48 gene as the most prevalent gene in 74% of isolates at the King Faisal Specialist Hospital and Research Center (2017–2018), and CAZ-AVI therapy showed a clinical cure rate of 80% [64]. Although both later studies depicted higher clinical success than our findings, these domestic reports did not highlight any independent variables (predictors) that might have influenced their clinical or microbiological success findings. Furthermore, our study provides in-depth clinical insights for CAZ-AVI applications over wider regimens as it solely analyzes the differential clinical/microbial outcomes between the studied groups on the basis of antibiotic regimen, monotherapy, and add-on therapy. Finally, the study by Alraddadi et al. utilized a very small sample size (CAZ-AVI-treated patients; *n* = 10) in a way that could greatly influence its findings.

Previous studies have illustrated varied mortality rates for carbapenem-resistant *Enterobacterales* infections, ranging from 8.6% to up to 50% within different populations [34,58,65,66]. In their study, Nagvekar et al. found that the utilization of CAZ-AVI as a standalone treatment or in conjunction with other medications resulted in a substantial success rate among patients. Furthermore, the data revealed an overall mortality rate of 21%, indicating that CAZ-AVI may be a promising therapeutic choice for individuals afflicted with carbapenem-resistant *Enterobacterales* infections [67]. Zhang et al. found that patients who were treated with CAZ-AVI had a much lower chance of dying within 30 days than those who received other treatments for carbapenem-resistant *K. pneumonia* infections after kidney transplantation [61]. Tumbarello et al. found that the 30-day mortality rate of 104 patients with bacteremic *K. pneumonia* carbapenemase-producing *K. pneumonia* infections was significantly lower than that of a matched cohort whose KPC-Kp bacteremia had been treated with drugs other than CAZ-AVI (36.5% vs. 55.8%) [31]. Several independent factors were identified as significant influencers of the 30-day mortality rates; these include pneumonia, the hospitalization length of stay, baseline creatinine clearance, the Charlson comorbidity score, an INCREMENT index above or equal to eight, obesity, CAZ-AVI renal dose adjustment, prolonged CAZ-AVI infusion, and/or septic shock [34,58,63]. Chen et al. performed a meta-analysis on how well CAZ-AVI worked and how safe it was for treating carbapenem-resistant enterobacterial bloodstream infections from the extracted data of 11 studies and large combined patient sample sizes (*n* = 1205) [68]. The study found that patients in the CAZ-AVI group had a much lower 30-day death rate than those on colistin-based regimens. A comparative study between CAZ-AVI and polymyxin B highlighted higher CAZ-AVI’s bacterial clearance (~43% versus 14%) and lower 30-day mortality (~14% versus 43%) [69]. In line with previous studies, our observations recapitulate the literature-reported evidence, as our findings illustrated lower 30-day mortality rates for CAZ-AVI against comparative standard therapy, with sex (female %), ICU admission, and fever as negative predictors. However, the administration of CAZ-AVI within our investigated intervention group itself was just a 50:50 chance for treatment success compared to treatment failure (32 versus 32 CAZ-AVI-treated patients). Only the study by Alraddadi et al. reported insignificant mortality differences among CAZ-AVI-treated inpatients compared to those with tigecycline- or polymyxin-containing regimens [64]. However, the small sample size could be the reason behind such a finding. In summary, patient mortality and drug regimen efficacy are generally associated with a patient’s characteristics, baseline features, and drug-related factors. 

Finally, the presented study was limited by the retrospective nature of the medium-sized sample with relatively complex co-morbidities. Selection bias could not be entirely ruled out, as the study design was not blind in terms of the fact that the investigator did not know which treatment regimen was being used or observe which treatment regimen was more effective when combined with CAZ-AVI. The study further lacked more detailed information regarding infection severity indicators, patients’ renal/liver status, and lab analysis to assess the pharamcokinetic aspects in terms of efficiency. Owing to its limited sample size, we could further sub-analyze OXA-48-like variants with or without other co-producing β-lactamase-resistant mutations. Other empirical antibiotics prescribed prior to OXA-48-like antibiotics could impact drug efficiency and mortality, which should not be ignored. More research regarding multicenter prospective large studies is recommended to determine the CAZ-AVI clinical efficiency and which suitable antibiotic regimen should be added to CAZ-AVI. 

## 4. Materials and Methods

### 4.1. Ethical Statements and Study Design

The present study is a retrospective cohort single-center study that was conducted as per the WMA Declaration of Helsinki Ethical Standards as well as being revised and approved by the Research Ethics Committee board of the institute (#AFHSRMREC/2022/MICROBIOLOGY/661). This study was exempt from the requirements of patients’ inscribed informed consent owing to its observational and retrospective nature. Moreover, the Research Ethics Committee board approved the study and waived the need for such consents. This study involved patients who were admitted to the Armed Forces Hospitals, Saudi Southern Region, Khamis Mushait, Saudi Arabia, from 1 August 2018 until 1 May 2023, providing a comparative evaluation between the two groups: *Group I,* the control patients who received standard antibiotic therapy for the treatment of *K. pneumonia* infection and considered antibiotics including carbapenem, colistin, tigecycline, and aminoglycoside antibiotics; *Group II*, patients who received CAZ-AVI alone and add-ons to the standard antibiotics. The groups were compared on the basis of clinical outcomes in terms of 30 days, all of which cause mortality, clinical remission, microbial recurrence, and eradication.

The Armed Forces Hospital in the southern region is regarded as one of the greatest medical organizations in the world, providing medical and health services (curative, preventive, and diagnostic) in all medical specialties to employees of the Ministry of Defense in the southern region. This medical complex serves as a reference hospital for critical cases transferred by military hospitals from Sharurah, Najran, and Jizan, as well as cases received from civilian hospitals within the region. Over the course of four decades, hospital facilities were developed and expanded to become today’s medically integrated monument in the southern region, where the bed capacity was increased to 571 beds, serving as an integrated hospital center for the near future.

### 4.2. Patients’ Inclusion and Exclusion Criteria

This study included the following: (1) admitted patients aged more than 18 years old in any department of the hospital, including ICU-admitted patients and patients transferred to the ICU for treatment; (2) patients with *K. pneumonia* OXA 48-like genes with any site of infection and being confirmed through drug sensitivity and bacterial cultures; and (3) patients who received one or more antibiotic treatment starting from positive culture results and for at least five days. The exclusion criteria were as follows: (1) patients aged younger than 18 years old; (2) patients who died prior to receiving the OXA-48 antibiotic therapy or who could not be assessed for clinical efficiency (who used antibiotics for less than 5 days); and (3) patients who lacked the OXA-48-like gene for *K. pneumonia*. 

### 4.3. Data Collection

The clinical data of the included patients were obtained using the data collection sheets, which were followed up till discharge or death. These forms included the patient’s demographic characteristics (age and sex) in addition to the site of infections, baseline co-morbidity, laboratory results, ICU stays, details of the antibiotics received, and clinical status of the patient. The data were collected by two clinical pharmacists and a microbiology lab technologist, and they were then anonymized. Infectious disease specialists in the Armed Forces Hospital Southern Region evaluated the collected data in terms of clinical success, microbial eradication, and recurrence. These outcomes were evaluated according to the definitions given below.

The study’s primary outcome was 30-day all-cause mortalities, which were defined as all-cause deaths that happened within 30 days after the bacterial isolates obtained from the infected patients (first-positive cultures for the OXA-48-like gene in CRKP patients) [42]. The CRKP isolates were defined as per the Clinical and Laboratory Standards Institute (CLSI) breakpoints, where the minimum inhibitory concentrations (MICs) for imipenem or meropenem were equal to or more than 4 mg/L [70]. Secondary outcomes included clinical success and antibiotic-microbiological efficiency indices such as bacterial recurrence and microbial eradication. Clinical success was defined as improvement in the symptoms and signs starting from baseline and till therapy end, including the following: defervescence (fever remission; <38.0 °C or 100.4 °F for 48 h), normalization of WBC counts (<11.0 × 10^9^ /L), procalcitonin (≤0.05 μg/L), and/or C-reactive protein (≤3 mg/L) blood levels [43,44]. Microbial eradication was defined as obtaining two consecutive negative cultures from the same and/or different sites [45]. Bacterial recurrence/relapse was defined as bacteremia with the same species or susceptibility pattern obtained from the blood isolates, following at least one negative microbe growth [44]. 

### 4.4. Data Analysis

The statistical analyses were conducted using open-source JASP^®^ v.0.18.3 statistical software with a dynamic update of results (University of Amsterdam, Amsterdam, The Netherlands) and GraphPad Prism^®^ v5.01 (La Jolla, CA, USA) that were also used for the analysis and graphical representations whenever appropriate. A significance level of *p* < 0.05 was considered to indicate statistical significance. The demographic data were analyzed using descriptive statistics (medians with interquartile ranges, medians with minimum–maximum ranges, or means ± standard error of means), and the findings were reported in numerical and percentage formats. The investigation focused on determining the superiority of each treatment arm by applying an independent sample *t*-test or Mann–Whitney U test for normally or non-normally distributed data, respectively. In regard to the categorical data, they are expressed in case numbers and percentages and then analyzed using contingency testing. Univariate and multivariable logistic regression analyses were used to assess any potential CAZ-AVI’s clinical and microbiological efficiency-associated independent variables (predictors), including patients’ demography, ICU admission, inotropes administration, infection sites, co-morbidities, details of antibiotic usage, clinical signs, and lab results.

## 5. Conclusions

This study demonstrates a promising result of CAZ-AVI add-on therapy compared to standard antibiotic regimens for treating infections caused by carbapenem-resistant *K. pneumonia* with bla OXA-48-like genes in regard to clinical remission, microbial eradication, reducing the bacterial recurrence, and reducing 30-day all-cause hospital mortality. The impact of different independent variables like sex, fever, ICU admissions, inotropes, CRP bio levels, the duration of CAZ-AVI therapy, and/or WBC counts on CAZ-AVI’s clinical and microbiological outcomes was also highlighted. Clinicians should individualize the CAZ-AVI dose based on co-existing risk factors to achieve optimal efficacy. Prospective multicenter and randomized control studies are recommended with individualized CAZ-AVI precision administrations based on patients’ different sites of infection, bacterial resistance mechanisms, renal/liver function, monitored blood concentration, and others.

## Figures and Tables

**Figure 1 antibiotics-13-00265-f001:**
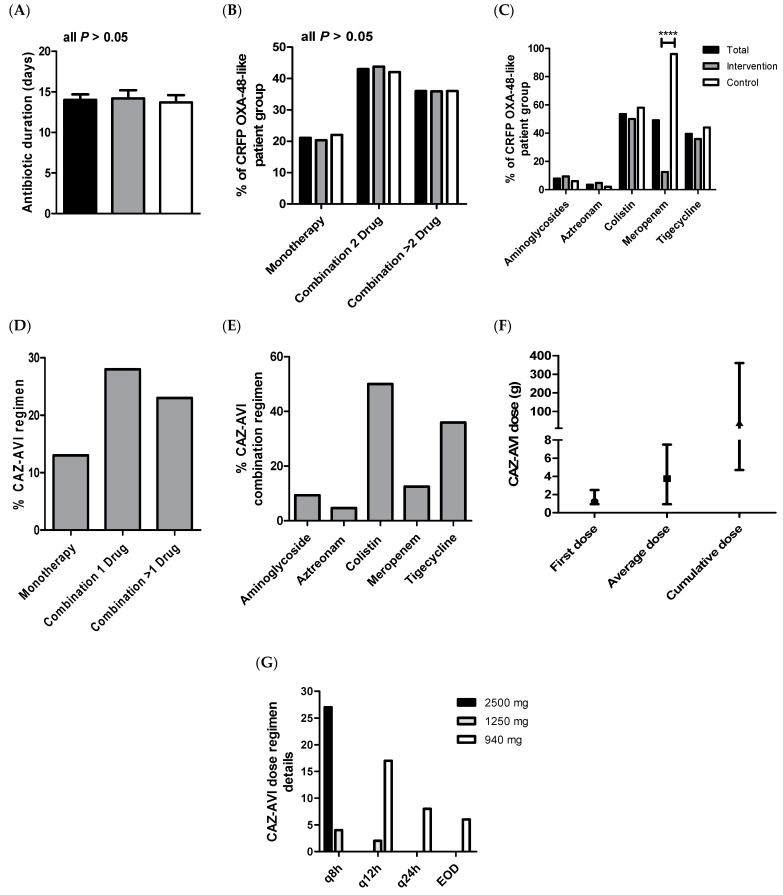
Antibiotic drug usage within CRKP OXA-48-like patients. (**A**) Antibiotic duration of treatment in days. (**B**) Antibiotic sole and combination regimens. (**C**) Percentages of antibiotic members across total, intervention, and control patient groups. (**D**) CAZ-AVI sole and combination regimens in intervention patients. (**E**) Percentages of CAZ-AVI combined with antibiotic members of the standard therapy. (**F**) Dose regimen details of CAZ-AVI usage (median with maximum and minimum ranges). (**G**) Frequency of the CAZ-AVI dose regimen as per patients’ physiological status. Comparative antibiotic data between intervention versus the control groups were statistically insignificant for almost all items (*p*-values > 0.05). **** *p* < 0.0001.

**Figure 2 antibiotics-13-00265-f002:**
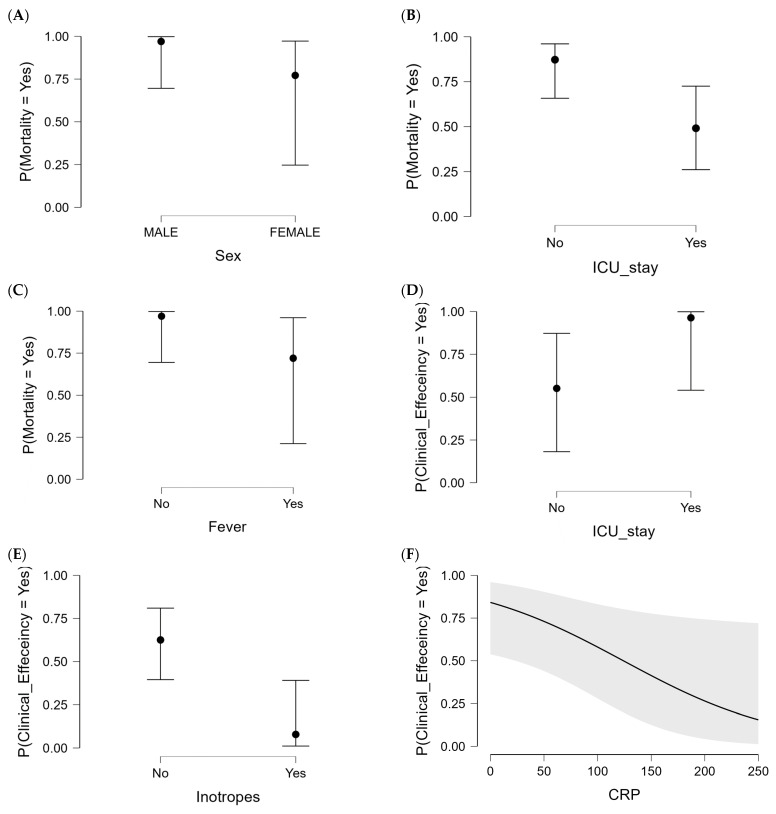
Conditional estimation/prediction plots for the multivariate logistic regression analyses between CAZ-AVI outcomes and independent variables (line equation) at 95% confidence intervals (shadow part). (**A**–**C**) The 30-day all-cause mortality. (**D**–**G**) Clinical efficiency. (**H**–**J**) Microbial eradication. Only parameters that were significant (*p* < 0.05) throughout the multivariate logistic regression analyses were represented.

**Table 1 antibiotics-13-00265-t001:** Demographic features and baseline clinical characteristics of patients with OXA-48-like CRKP infections.

Variables	Total Admitted Patients (*n* = 114)	Intervention (*n* = 64)	Control (*n* = 50)	*p*-Values
Age (years) *	71 (20.0–102.0)	75 (20.0–102.0)	69 (27.0–97.0)	0.094
Sex (Female)	48 (42.1%)	25 (39.1%)	23 (46.0%)	0.556
ICU admissions	85 (74.6%)	**42 (65.6%)**	**43 (86.0%)**	**0.017**
Inotropes **	56 (49.1%)	27 (42.2%)	29 (58.0%)	0.131
**Sites of infections**				
Multi-site infection ***	28 (24.6%)	11 (17.20%)	17 (34.0%)	0.663
Bloodstream	30 (26.3%)	**18 (28.1%)**	**12 (24.0%)**	**0.053**
Respiratory tract	31 (27.2%)	14 (21.9%)	17 (34.0%)	1.000
Soft tissues	21 (18.4%)	18 (28.1%)	3 (6.0%)	0.003
Urinary tract	4 (3.5%)	3 (4.7%)	1 (2.0%)	0.318
**Co-morbidities**				
Respiratory diseases	37 (32.5%)	20 (31.3%)	17 (34.0%)	0.841
Cardiovascular diseases	83 (72.8%)	46 (71.9%)	37 (74.0%)	0.835
Diabetes mellitus	69 (60.5%)	40 (62.5%)	29 (58.0%)	0.701
Kidney diseases	46 (40.4%)	26 (40.6%)	20 (40.0%)	1.000
Central nervous system diseases	16 (14.0%)	8 (12.5%)	8 (16.0%)	0.594
Cerebrovascular diseases	23 (20.2%)	10 (15.6%)	13 (26.0%)	0.240
Gastrointestinal diseases	5 (4.4%)	3 (4.7%)	2 (4.0%)	1.000
Septic shock/sepsis	29 (25.4%)	15 (23.4%)	14 (28.0%)	0.666
Tumors	2 (1.7%)	1 (1.6%)	1 (2.0%)	1.000
COVID-19 infections	21 (18.4%)	13 (20.3%)	8 (16.0%)	0.631
**Antibiotic usage**				
Duration time of treatment (days) *	14.0 (±0.7)	14.2 (±1.0)	13.7 (±0.9)	0.881
Monotherapy	24 (21.1%)	13 (20.3%)	11 (22.0%)	0.825
Combinations of two agents	49 (43.0%)	28 (43.8%)	21 (42.0%)	1.000
Combinations of ≥triple agents	41 (36.0%)	23 (35.9%)	18 (36.0%)	1.000
Aminoglycosides	9 (7.9%)	6 (9.4%)	3 (6.0%)	0.729
Aztreonam	4 (3.5%)	3 (4.7%)	1 (2.0%)	0.630
Colistin	61 (53.5%)	32 (50.0%)	29 (58.0%)	0.452
Meropenem	56 (49.1%)	**8 (12.5%)**	**48 (96.0%)**	**<0.0001**
Tigecycline	45 (39.5%)	23 (35.9%)	22 (44.0%)	0.442
**Lab and clinical signs**				
WBC counts (×10^9^/L) *	12.0 (±0.7)	11.3 (±0.7)	13.0 (±1.2)	0.355
Neutrophil counts (×10^9^/L) *	12.7 (±2.0)	12.7 (±2.1)	12.6 (±1.9)	0.261
C-reactive protein (mg/L) *	121.0 (±11.7)	**80.1 (±8.7)**	**169.3 (±21.4)**	**0.0011**
Fervescence ****	62 (54.4%)	**27 (42.2%)**	**35 (70.0%)**	**0.004**

Otherwise undefined, data are represented as case numbers and their percentages out of total. * Data representation as per median (minimum–maximum) or mean (± standard error of mean). ** Inotropes were administered vasoactive agents including dopamine, epinephrine, norepinephrine, and/or vasopressin. *** Multi-site infections; infections via the OXA-48-like CRKP strain at more than one site. **** Temperatures of 38.0 °C (100.4 °F) or above for 48 h were considered fever. *p*-values were estimated using a *t*-test, Mann–Whitney U test, or contingency testing (Chi-square or Fisher’s exact test) based on the data. Values in bold represent statistical significance (*p* < 0.05).

**Table 2 antibiotics-13-00265-t002:** Clinical and microbiological outcomes of patients with OXA-48-like CRKP infections.

Outcomes *	Total Admitted Patients (*n* = 114)	Intervention (*n* = 64)	Control (*n* = 50)	*p*-Values
Clinical success	38 (33.3%)	27 (42.2%)	11 (22.0%)	**0.028**
Microbial eradication	65 (57.0%)	44 (68.8%)	21 (42.0%)	**0.007**
Bacterial recurrence	17 (14.9%)	5 (7.8%)	12 (24.0%)	**0.019**
30-day all-cause mortality	67 (58.7%)	32 (50.0%)	35 (70.0%)	**0.036**
	**Total** **Monotherapy** **Patients (*n* = 24)**	**Intervention** **Monotherapy** **(*n* = 13)**	**Control** **Monotherapy** **(*n* = 11)**	***p*-** **Values**
Clinical success	9 (37.5%)	6 (46.2%)	3 (27.3%)	0.423
Microbial eradication	14 (58.3%)	10 (76.9%)	4 (36.4%)	0.095
Bacterial recurrence	7 (29.2%)	3 (23.1%)	4 (36.4%)	0.659
30-day all-cause mortality	15 (62.5%)	9 (69.2%)	6 (54.5%)	0.675
	**Total** **Combined therapy** **Patients (*n* = 90)**	**Intervention** **Add-on Therapy** **(*n* = 51)**	**Control** **Combined Therapy** **(*n* = 39)**	** *p* ** **-Values**
Clinical success	29 (32.2%)	21 (41.2%)	8 (20.5%)	**0.043**
Microbial eradication	51 (56.7%)	34 (66.7%)	17 (43.6%)	**0.034**
Bacterial recurrence	10 (11.11%)	2 (3.9%)	8 (20.5%)	**0.018**
30-day all-cause mortality	52 (57.8%)	23 (45.1%)	29 (74.4%)	**0.009**

Data are represented as case numbers and their percentages out of the total or respective patient group. ***** Bacterial recurrence = bacteremia with the same species and susceptibility pattern as the index blood isolate, following at least one negative microbe growth. Clinical success = fever remission, plus normalization of WBC count, procalcitonin, and C-reactive protein. Microbial eradication = two consecutive negative cultures from the same and different sites. The 30-day all-cause mortality = any cause of death that happened within 30 days of bacterial isolates. p-values were estimated through contingency testing (Chi-square or Fisher’s exact test) based on the data. Values in bold represent statistical significance (*p* < 0.05).

**Table 3 antibiotics-13-00265-t003:** Variables and risk factor analysis for clinical success/efficacy and 30-day all-cause mortality of CAZ-AVI-driven antibiotic regimens within CRKP OXA-48-like infected patients.

Variables	Clinical Efficiency			30-Day All-Cause Mortality		
	CAZ-AVI Treatment Success (*n* = 27)	CAZ-AVI Treatment Failure (*n* = 37)	*p*-Values	CAZ-AVI Patient Survived (*n* = 32)	CAZ-AVI Patient Deceased (*n* = 32)	*p*-Values
Age (years) *	71 (27.0–97.0)	67 (31.0–93.0)	0.960	69 (27.0–97.0)	70 (39.0–95.0)	0.333
Sex (Female)	**7 (25.9%)**	**18 (48.6%)**	**0.026**	**8 (25.0%)**	**17 (53.1%)**	**0.039**
ICU admissions	**13 (48.1%)**	**29 (78.4%)**	**0.017**	**14 (43.8%)**	**28 (87.5%)**	**0.017**
Inotropes	**4 (14.8%)**	**23 (62.2%)**	**<0.001**	**4 (12.5%)**	**23 (71.9%)**	**<0.0001**
**Sites of infections**						
Multi-site infection	5 (18.5%)	6 (16.2%)	1.000	7 (21.9%)	4 (12.5%)	0.509
Bloodstream	6 (22.2%)	12 (32.4%)	0.413	8 (25%)	10 (31.3%)	0.782
Respiratory tract	5 (18.5%)	9 (24.3%)	0.761	6 (18.8%)	8 (25.0%)	0.763
Soft tissues	**12 (44.4%)**	**6 (16.2%)**	**0.023**	**13 (40.6%)**	**5 (15.6%)**	**0.049**
Urinary tract	1 (3.7%)	2 (5.4%)	1.000	0 (0.0%)	3 (9.4%)	0.238
**Co-morbidities**						
Respiratory diseases	8 (29.6%)	12 (32.4%)	1.000	8 (25.0%)	12 (37.5%)	0.419
Cardiovascular diseases	21 (77.8%)	25 (67.6%)	0.413	22 (68.8%)	24 (75.0%)	0.782
Diabetes mellitus	17 (63.0%)	23 (62.1%)	1.000	18 (56.3%)	22 (68.8%)	0.439
Kidney diseases	11 (40.7%)	15 (40.5%)	1.000	12 (37.5%)	14 (43.75%)	0.799
Central nervous system diseases	3 (11.1%)	5 (13.5%)	1.000	4 (12.5%)	4 (12.5%)	1.000
Cerebrovascular diseases	4 (14.8%)	6 (16.2%)	1.000	4 (12.5%)	6 (18.8%)	0.732
Gastrointestinal diseases	1 (3.7%)	2 (5.4%)	1.000	1 (3.1%)	2 (6.3%)	1.000
Septic shock/sepsis	4 (14.8%)	11 (29.7%)	0.235	**3 (9.4%)**	**12 (37.5%)**	**0.016**
Tumors	0 (0.0%)	1 (2.7%)	1.000	0 (0.0%)	1 (3.1%)	1.000
COVID-19 infections	5 (18.5%)	8 (21.6%)	1.000	3 (9.4%)	10 (31.3%)	0.060
**Antibiotic usage**						
CAZ-AVI duration therapy (days) *	**16.8 (±1.7)**	**11.6 (±1.0)**	**0.003**	16.1 (±1.6)	11.4 (±0.8)	0.037
CAZ-AVI monotherapy	6 (22.2%)	7 (18.9%)	0.763	5 (15.6%)	8 (25%)	0.536
Combinations of two agents	13 (48.1%)	15 (40.5%)	0.615	16 (50.0%)	12 (37.5%)	0.450
Combinations of ≥ triple agents	10 (37.0%)	13 (35.1%)	1.000	13 (40.6%)	10 (31.3%)	0.603
Aminoglycosides	2 (7.4%)	4 (10.8%)	1.000	5 (15.6%)	1 (3.1%)	0.105
Aztreonam	1 (3.7%)	2 (5.4%)	1.000	1 (3.1%)	2 (6.3%)	1.000
Colistin	13 (50.0%)	19 (51.4%)	1.000	14 (43.8%)	18 (56.3%)	0.454
Meropenem	3 (48.1%)	5 (13.5%)	1.000	5 (15.6%)	3 (9.4%)	0.708
Tigecycline	12 (44.4%)	11 (29.7%)	0.294	11 (34.4%)	12 (37.5%)	1.000
CAZ-AVI first dosage (g)	2.5 (0.94–2.5)	0.94 (0.94–2.5)	0.223	2.5 (0.94–2.5)	0.94 (0.94–2.5)	0.023
CAZ-AVI average dosage (g/day)	4.4 (1.9–7.5)	2.2 (0.94–7.5)	0.100	7.5 (0.94–7.5)	2.2 (0.94–7.5)	0.016
CAZ-AVI cumulative dosage (g)	**60.0 (9.4–360.0)**	**37.1 (4.7–157.5)**	**0.002**	**66.7 (6.6–360.0)**	**29.7 (4.7–157.5)**	**<0.001**
**Lab and clinical signs**						
WBC counts (×10^9^/L) *	**7.6 (±0.5)**	**13.5 (±1.2)**	**<0.001**	**8.5 (±0.7)**	**13.1 (±1.3)**	**0.016**
Neutrophil counts (×10^9^/L) *	**7.12 (±2.1)**	**15.0 (±2.9)**	**<0.001**	**9.2 (±2.1)**	**13.3 (±2.7)**	**0.007**
C-reactive protein (mg/L) *	**25.1 (±5.6)**	**112.0 (±9.7)**	**<0.0001**	**47.5 (±9.4)**	**108.2 (±11.5)**	**<0.0001**
Fervescence	**2 (7.4%)**	**25 (67.6%)**	**<0.0001**	**4 (12.5%)**	**23 (71.9%)**	**<0.0001**

Otherwise undefined, data are represented as case numbers and their percentages out of total. * Data representation as per median (minimum–maximum) or mean (±standard error of mean). *p*-values were estimated through a *t*-test, Mann–Whitney U test, or contingency testing (Chi-square or Fisher’s exact test) based on the data. Values in bold represent statistical significance (*p* < 0.05).

**Table 4 antibiotics-13-00265-t004:** Variables and risk factor analysis for microbial eradication and bacterial recurrence/relapse of CAZ-AVI-driven antibiotic regimens within CRKP OXA-48-like infected patients.

Variables	Microbial Eradication			Bacterial Recurrence		
	CAZ-AVI Infection Eradicated (*n* = 44)	CAZ-AVI Infection Persistent (*n* = 20)	*p*-Values	CAZ-AV Infection Relapsed (*n* = 5)	CAZ-AVI Infection Receded (*n* = 59)	*p*-Values
Age (years) *	69 (27.0–97.0)	70 (39.0–95.0)	0.300	69 (49.0–77.0)	70 (27.0–97.0)	0.582
Sex (Female)	**8 (18.2%)**	**17 (85.0%)**	**0.011**	0 (0.0%)	25 (42.4%)	0.147
ICU admissions	**24 (54.5%)**	**18 (90.0%)**	**0.009**	2 (40.0%)	40 (67.8%)	0.329
Inotropes	**11 (25.0%)**	**16 (80.0%)**	**<0.0001**	0 (0.0%)	27 (45.8%)	0.068
**Sites of infections**						
Multi-site infection	8 (18.2%)	3 (15.0%)	1.000	2 (40.0%)	9 (15.3%)	0.201
Bloodstream	10 (22.7%)	8 (40.0%)	0.230	2 (40.0%)	16 (27.1%)	0.615
Respiratory tract	10 (22.7%)	4 (20.0%)	1.000	1 (20.0%)	13 (22.0%)	1.000
Soft tissues	15 (34.1%)	3 (15.0%)	0.143	0 (0.0%)	18 (30.5%)	0.310
Urinary tract	2 (4.5%)	1 (5.0%)	1.000	0 (0.0%)	3 (5.1%)	1.000
**Co-morbidities**						
Respiratory diseases	**10 (22.7%)**	**10 (50.0%)**	**0.042**	0 (0.0%)	20 (33.9%)	0.314
Cardiovascular diseases	32 (72.7%)	14 (70.0%)	1.000	4 (80.0%)	42 (71.2%)	1.000
Diabetes mellitus	29 (65.9%)	11 (55.0%)	0.419	2 (40.0%)	38 (64.4%)	0.355
Kidney diseases	17 (38.6%)	9 (45.0%)	0.784	2 (40.0%)	24 (40.7%)	1.000
Central nervous system diseases	6 (13.6%)	2 (10.0%)	1.000	1 (20.0%)	7 (11.9%)	0.499
Cerebrovascular diseases	7 (15.9%)	3 (15.0%)	0.728	1 (20.0%)	9 (15.3%)	1.000
Gastrointestinal diseases	1 (2.3%)	2 (10.0%)	0.228	0 (0.0%)	3 (5.1%)	1.000
Septic shock/sepsis	7 (15.9%)	8 (40.0%)	0.207	0 (0.0%)	15 (25.4%)	0.329
Tumors	0 (0.0%)	1 (5.0%)	0.313	0 (0.0%)	1 (1.7%)	1.000
COVID-19 infections	**4 (9.1%)**	**9 (45.0%)**	**0.002**	0 (0.0%)	13 (22.0%)	0.574
**Antibiotic usage**						
CAZ-AVI duration therapy (days) *	**16.0 (±1.2)**	**8.7 (±0.8)**	**<0.0001**	17.4 (±2.5)	13.4 (±1.0)	0.100
CAZ-AVI Monotherapy	10 (22.7%)	3 (15.0%)	0.739	0 (0.0%)	13 (22.0%)	0.574
Combinations of two agents	20 (45.5%)	8 (40.0%)	0.789	4 (80.0%)	24 (40.7%)	0.159
Combinations of ≥triple agents	15 (34.1%)	8 (40.0%)	0.780	1 (20.0%)	22 (37.3%)	0.646
Aminoglycosides	5 (11.4%)	1 (5.0%)	0.656	1 (20.0%)	5 (8.5%)	0.399
Aztreonam	1 (2.3%)	2 (10.0%)	0.228	0 (0.0%)	3 (5.1%)	1.000
Colistin	21 (47.7%)	11 (55.0%)	0.788	3 (60.0%)	29 (49.2%)	1.000
Meropenem	6 (13.6%)	2 (10.0%)	1.000	0 (0.0%)	8 (13.6%)	1.000
Tigecycline	16 (36.4%)	7 (35.0%)	1.000	2 (40.0%)	21 (35.6%)	1.000
CAZ-AVI first dosage (g)	1.25 (0.94–2.5)	0.94 (0.94–2.5)	0.217	2.5 (0.94–2.5)	1.25 (0.94–2.5)	0.346
CAZ-AVI average dosage (g/day)	3.8 (0.94–7.5)	2.2 (0.94–7.5)	0.133	5.0 (1.9–7.5)	3.8 (0.94–7.5)	0.434
CAZ-AVI cumulative dosage (g)	**54.4 (6.1–360.0)**	**19.4 (4.7–127.5)**	**<0.001**	105.0 (18.8–127.5)	39.5 (4.7–360.0)	0.080
**Lab and clinical signs**						
WBC counts (×10^9^/L) *	**8.2 (±0.5)**	**16.5 (±1.6)**	**<0.0001**	7.1 (±0.7)	11.1 (±0.8)	0.154
Neutrophil counts (×10^9^/L) *	**8.0 (±1.8)**	**18.2 (±4.0)**	**<0.0001**	**3.9 (±0.7)**	**11.9 (±2.0)**	**0.035**
C-reactive protein (mg/L) *	**59.8 (±9.8)**	**114.3 (±11.7)**	**<0.001**	**21.1 (±9.6)**	**80.9 (±8.7)**	**0.025**
Fervescence	**9 (20.5%)**	**18 (90.0%)**	**0.017**	1 (20.0%)	26 (44.1%)	0.387

Otherwise undefined, data are represented as case numbers and their percentages out of total. * Data representation as per median (minimum–maximum) or mean (±standard error of mean). *p*-values were estimated through a *t*-test, Mann–Whitney U test, or contingency testing (Chi-square or Fisher’s exact test) based on the data. Values in bold represent statistical significance (*p* < 0.05).

**Table 5 antibiotics-13-00265-t005:** Multivariate logistic regression analysis of clinical and microbiological outcomes within CAZ-AVI-treated CRKP OXA-48-like infected patients.

**Variables ***	**30-Day All-Cause Mortality**				**Clinical Efficiency**			
	**B**	**ORs (95% CI)**	***p*-Values**	**VIF (TI)**	**B**	**ORs (95% CI)**	***p*-Values**	**VIF (TI)**
Sex	**−2.252**	**0.105 (–4.200; −0.305)**	**0.023**	**1.78 (0.56)**	−1.598	0.202 (–4.229; 1.033)	0.234	1.95 (0.51)
ICU admissions	**−1.957**	**0.141 (–3.518; −0.397)**	**0.014**	**1.45 (0.69)**	**3.053**	**21.183 (–0.102; 6.208)**	**0.048**	**3.13 (0.32)**
Inotropes	−0.801	0.449 (–2.827; 1.223)	0.438	1.92 (0.52)	**−2.974**	**0.051 (–4.795; −1.153)**	**0.001**	**1.91 (0.53)**
Soft tissues	0.002	1.002 (–2.166; 2.169)	0.999	1.73 (0.58)				
Septic shock/sepsis	−0.750	0.473 (–2.970; 1.470)	0.508	1.55 (0.65)				
CAZ-AVI duration					0.049	0.952 (–0.219; 0.121)	0.572	1.70 (0.59)
CAZ-AVI cumulative	0.005	1.005 (–0.013; 0.024)	0.563	1.31 (0.77)	0.008	1.008 (–0.020; 0.036)	0.583	1.93 (0.52)
WBC counts	0.005	1.005 (–0.164; 0.175)	0.951	1.52 (0.66)	−0.235	0.790 (–0.564; 0.093)	0.160	1.73 (0.58)
Neutrophil counts	0.030	1.031 (–0.025; 0.085)	0.282	1.27 (0.79)	−0.020	0.981 (–0.125; 0.086)	0.715	1.33 (0.75)
C-reactive protein	−0.002	0.998 (–0.016; 0.012)	0.766	1.85 (0.54)	**−0.013**	**0.987 (–0.026; −0.000)**	**0.043**	**1.81 (0.55)**
Fervescence	**−2.524**	**0.080 (–4.465; −0.582)**	**0.011**	**1.70 (0.59)**	**−5.596**	**0.004 (–9.507; −1.685)**	**0.004**	**2.46 (0.41)**
**Variables ***	**Microbial Eradication**					**Bacterial Recurrence**		
	**B**	**ORs (95% CI)**	***p*-Values**	**VIF (TI)**	**B**	**ORs (95% CI)**	***p*-Values**	**VIF (TI)**
Sex	−2.866	0.057 (–6.318; 0.586)	0.104	3.50 (0.29)				
ICU admissions	−1.952	0.142 (–4.441; 0.536)	0.124	1.08 (0.93)				
Inotropes	−1.620	0.198 (–3.434; 0.195)	0.080	1.36 (0.74)				
Respiratory diseases	1.373	3.949 (–0989; 3.736)	0.255	1.56 (0.64)				
COVID-19 infections	0.476	1.609 (–2.066; 3.018)	0.714	1.47 (0.68)				
CAZ-AVI duration	**0.368**	**1.446 (0.101; 0.636)**	**0.007**	**1.26 (0.79)**				
CAZ-AVI cumulative	0.002	1.002 (–0.028; 0.031)	0.920	1.32 (0.76)				
WBC counts	**−0.292**	**0.747 (–0.563; −0.021)**	**0.034**	**1.48 (0.67)**				
Neutrophil counts	−0.004	0.996 (–0.063; 0.055)	0.903	1.34 (0.75)	−0.298	0.742 (–0.842; 0.247)	0.284	1.05 (0.95)
C-reactive protein	−0.004	0.996 (–0.018; 0.010)	0.598	1.09 (0.92)	−0.022	0.978 (–0.060; 0.016)	0.257	1.05 (0.95)
Fervescence	**−4.353**	**0.013 (–7.499; −1.206)**	**0.007**	**2.75 (0.36)**				

* Only variables depicted significance (*p* < 0.05) throughout the univariate analysis were included within the multivariate logistic regression analysis. B = regression coefficient; ORs = odds ratios; CI = confidence intervals. Values in bold represent statistical significance (*p* < 0.05).

## Data Availability

Data supporting reported results can be requested by contacting the corresponding author.

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
