# Peer review of "The Clinical Efficacy of Adding Ceftazidime/Avibactam to Standard Therapy in Treating Infections Caused by Carbapenem-Resistant Klebsiella pneumonia with blaOXA-48-like Genes"

_antibiotics, 2024, doi:10.3390/antibiotics13030265_

Round 1

Reviewer 1 Report (Previous Reviewer 3)

Comments and Suggestions for Authors

The authors describe a retrospective analysis of the treatment with ceftazidime/avibactam of severe patients infected with enterobacteriales-producing oxa 48 such as cabapenemases.

There is a paucity of data on this topic and this analysis is important.

The manuscript is potentially flawed and should be clarified:

As far as I understand, the study referred to ceftazidime-avibactam being given empirically for infections secondary to blaOXA 48-like producers. If this is the case, it should be stated more clearly.

1-The treatment received in the CAZAVI and control groups is unclear:

You mentioned that CAZAVI is an add-on treatment, but at least part of the patients did not receive combination therapy?

In the CAZAVI arm: in Table 1, please detail the empirical therapy: CAZAVI alone, biotherapy with mero, tige, coli AG etc.... and the documented therapy similarly detailed.

In the control group, the antibiotics were similarly detailed.

Double carbapenem has also been proposed for blaOXA48 EB, do you use the combination of meropenem and ertapenem? Please clearly detail the empirical and secondarily documented therapy.

2- The dose of CAZAVI is lower than recommended. Usually the dose is 2.5g tid. Lin 375 you mentioned 950 to 2500 mg per day and a mean daily dose of 1.25g?!!! please give details about the renal clearance and for how many patients the doses received are in accordance with recommendations?

3- Summary: Please state the site of infection, the % of ICU patients in the summary. Please include the p-values between groups (e.g. 50% vs 70%, p=....) etc.

4- The overall clinical success rate is only 30%, this is not an acceptable clinical success rate...especially if the mortality rate is less than 70%, please explain. What is the rescue therapy in case of clinical failure? In both groups?

5- Soft tissue infections are largely unevenly distributed between the groups. Is this an unusual infection? A post-hoc analysis excluding SSTI would have been interesting.

6- The multivariate analyses and Table 5 and Figure 2 are beyond the scope of the article. They could be deleted. Duration of therapy should not be considered as a risk factor for success in a logistic regression (problem with early deaths = competing events).

7- Minor: Enterobacteriaceae should be replaced by its new name: Enterobacterale.

Author Response

Dear Respected Editor-in-Chief of Antibiotics

On behalf of the authors, I would like to thank you for your consideration of our manuscript entitled "The clinical efficacy of adding Ceftazidime/Avibactam to standard therapy in treating infections caused by Carbapenem-resistant Klebsiella pneumonia with bla OXA-48-like genes". Your comments and those of the reviewers were highly insightful and enabled us to enhance the quality of our manuscript.

The authors have thoroughly revised the manuscript and replied point-by-point to each of the recommendations of the reviewers.

The authors hope that the revisions in the manuscript and our accompanying responses will be sufficient to make our manuscript suitable for publication in your reputable journal “Antibiotics.

The authors shall look forward to hearing from you at your earliest convenience.

Please find the attached PDF file

Reviewer 2 Report (Previous Reviewer 1)

Comments and Suggestions for Authors

The authors have addressed all my comments.

Author Response

Authors thank the reviewer for time and efforts in reviewing the manuscript.

Round 2

Reviewer 1 Report (Previous Reviewer 3)

Comments and Suggestions for Authors

the authors answered to my questions

This manuscript is a resubmission of an earlier submission. The following is a list of the peer review reports and author responses from that submission.

Round 1

Reviewer 1 Report

Comments and Suggestions for Authors

The authors assessed the efficacy of adding Ceftazidime/Avibactam to standard therapy for treating infections caused by OXA-48-like carbapenemase-producing Klebsiella pneumonia. Ceftazidime/Avibactam (CAZ-AVI) plays a crucial role in the treatment of Carbapenem-Resistant Klebsiella pneumoniae (CRKP) infections, and evaluating its clinical effectiveness is essential for guiding treatment decisions. However, the study may have several limitations.

1.     The study compared clinical data between the CAZ-AVI addition group and the control group. According to the study's methodology, both groups received standard antibiotic therapies, including carbapenems, colistin, tigecycline, tigecycline, and aminoglycoside antibiotics. The difference in outcomes may be due to variations in standard antibiotic therapy. Therefore, the authors should consider categorizing patients into specific treatment subgroups, such as those receiving carbapenem+CAZ-AVI or colistin+CAZ-AVI.

2.     The study did not specify when CAZ-AVI was administered. Typically, the need for CAZ-AVI arises when standard antibiotic therapy fails or the patient's condition deteriorates. Consequently, the results may not accurately reflect the efficacy of CAZ-AVI, as the patient populations in the two groups differ.

3.     The manuscript contains numerous typographical errors, such as 'Table 65' in line 94 and 'class A carbapenemase genes K' in line 61. Capitalization throughout the text is inconsistent, and Latin names are not consistently italicized.

4.     I suggest that the authors combine Tables 2 and 3 into a single table, as the information presented appears to be duplicated.

Comments on the Quality of English Language

The English of the manuscript is fine.

Author Response

Dear Respected Editor-in-Chief of Antibiotics

On behalf of the authors, I would like to thank you for your consideration of our manuscript (antibiotics-2699924) entitled "The clinical efficacy of adding Ceftazidime/Avibactam to standard therapy in treating infections caused by Carbapenem-resistant Klebsiella pneumonia with bla OXA-48-like genes". Your comments and those of the reviewers were highly insightful and enabled us to enhance the quality of our manuscript.

The authors have thoroughly revised the manuscript and replied point-by-point to each of the recommendations of the reviewers. Here, the authors resubmit the revised version as being encouraged through the editorial decision.

Revisions in the text are shown as; cyan highlights are the responses for reviewer 1, Yellow highlights for reviewer 2, and green highlights are the responses for reviewer 3.

The authors hope that the revisions in the manuscript and our accompanying responses will be sufficient to make our manuscript suitable for publication in your reputable journal “Antibiotics.

The authors shall look forward to hearing from you at your earliest convenience.

Please find the attached PDF file of Author's Reply to the Review Report (Reviewer 1)

Reviewer 2 Report

Comments and Suggestions for Authors

In this manuscript, retrospectively, the authors determine the outcome of addition of CAZ-AVI to the standard therapy in treating infections caused by Klebsiella pneumoniae harboring OXA-48 resistant gene.

Klebsiella spelling needs to be fixed throughout the manuscript: For example, Line 31, 33, 48, 49, 73, 85.

Line 41, 75: Spelling "Gram-negative"

Line 42: CRE acronym needs to be corrected to Carbapenem- resistant Enterobacteriaceae.

Italicize Klebsiella throughout the manuscript.

Line 91: Please detail out the standard therapy with which the comparison has been made.

Line 94/95: Misplaced text. What does Table 65 mean?

Table 2 and table 3 headings read the same. It should mention/highlight what is the difference between the two?

Overall, the manuscript lacks significant findings and with the data that has been shown, it looks weak. The difference in analysis is not sufficiently detailed out. Lack of originality/innovation needs to be addressed.

Comments on the Quality of English Language

Scientific spelling errors throughout the manuscript.

Easy to read, otherwise. 

Only minor edits needed.

Author Response

Reviewer # 2: Comments and Suggestions

In this manuscript, retrospectively, the authors determine the outcome of addition of CAZ-AVI to the standard therapy in treating infections caused by Klebsiella pneumonia harboring OXA-48 resistant gene. The authors would like to thank the reviewer for the time and effort in revision. Kindly, find the detailed responses and the revised manuscript as per the reviewer’s addressed comments and suggestions.

  1. Klebsiella spelling needs to be fixed throughout the manuscript: For example, Line 31, 33, 48, 49, 73, 85.

Response 1: Authors thank the reviewer for valuable comment. Typos were corrected throughout the manuscript. Kindly refer to lines 4, 22, 46, and 54.

  1. Line 41, 75: Spelling "Gram-negative"

Response 2: Authors thank the reviewer for valuable comment. Typos were corrected throughout the manuscript written as “gram-negative” (as per CDC preferred usage). Kindly refer to Yellow highlights at lines 17, 50, 56, 101, and 333.

  1. Line 42: CRE acronym needs to be corrected to Carbapenem- resistant Enterobacteriaceae.

Response 3: Authors thank the reviewer for valuable comment. Typos were corrected as per reviewer suggestion. Kindly refer to Yellow highlights at line 116.

  1. Italicize Klebsiella throughout the manuscript.

Response 4: Authors thank the reviewer for valuable comment. Typos were corrected throughout the manuscript. Kindly refer to lines 4, 22, 46, and 54.

  1. Line 91: Please detail out the standard therapy with which the comparison has been made.

Response 5: Authors thank the reviewer for valuable comment. Standard therapy was monotherapy, combination of two agents, or combination of more than two antibiotics. These included; aminoglycosides for 9 (7.9%), aztreonam for 4 (3.5%), colistin for 61 (53.5%), and tigecycline for 45 (39.5%) of the total patients cases. Details of their frequencies among the investigated patients are thoroughly presented within Table 1 and Figure 1B-E.

  1. Line 94/95: Misplaced text. What does Table 65 mean?

Response 6: Authors thank the reviewer for valuable comment. Tables and their context within the manuscript have been revised and modified as per the conducted deep analysis including superiority of each treatment arm in terms of patient’s demography, characteristics, site of infection, co-morbidities, details of antibiotic usage, and clinical signs / specific lab values. Comparative analysis was done through applying independent sample t-test or Mann-Whitney U test for normally or non-normally distributed data, respectively. In regard to categorical data, comparisons are expressed in case numbers and percentages and then analyzed using the contingency testing. Multivariable logistic regression analysis was to follow for assessing any potential association of independent variables with CAZ-AVI’s clinical and microbiological efficiencies (clinical success, 30-day all-cause mortality, microbial eradication, bacterial recurrence). Kindly, refer to Tables 1-5.

  1. Table 2 and table 3 headings read the same. It should mention/highlight what is the difference between the two?

Response 7: Authors thank the reviewer for valuable comment. The authors have categorized the investigated patient of the two-arm treatment groups as per the details of patients’ demography, site of infection, co-morbidities, antibiotic usage details (duration, combination, name of added antibiotic and doses), and clinical sign / special lab results. All tables have been revised and modified as per the conducted deep analysis including superiority of each treatment arm through applying independent sample t-test or Mann-Whitney U test for normally or non-normally distributed data, respectively. In regard to categorical data, they are expressed in case numbers and percentages and then analyzed using the contingency testing. Multivariable logistic regression analysis was to follow for assessing any potential CAZ-AVI efficiency-associated independent variables (predictors). Kindly, refer to Tables 1-5.

  1. Overall, the manuscript lacks significant findings and with the data that has been shown, it looks weak. The difference in analysis is not sufficiently detailed out. Lack of originality/innovation needs to be addressed.

Response 8: Authors thank the reviewer for valuable comment. Authors have performed rigorous comparative analyses between the treatment arms (intervention versus control) as well as the treatment success versus failure arms within the intervention group itself. Comparisons were made on bases of patients’ demography, site of infection, co-morbidities, antibiotic usage details (duration, combination, name of added antibiotic and doses), and clinical sign / lab results. Interesting and relevant findings have been highlighted as follows; Administration of CAZ-AVI showed better clinical success and microbial eradication as well as less mortality rates and bacterial recurrence as compared to control standard therapy. CAZ-AVI’s clinical success was 42.2%, while as the intervention versus comparative groups showed decreased 30-day all-cause mortality (50.0% versus 70.0%) and infection recurrence (7.8% versus 24.0%), as well as substantially increased the rate of microbial eradication (68.8% versus 42.0%).ï‚·Comparative clinical success, microbial eradication, 30-day mortalities, and bacterial recurrence were dissected throughout the patients’ intervention group (add on CAZ-AVI therapy) and highlighted through statistical analysis on bases of patients’ demography, site of infection, co-morbidities, antibiotic usage details (duration, combination, name of added antibiotic and doses), and clinical sign / lab results. Authors further investigated through univariate and multivariate logistic regressions to identify the independent factors and covariates (predictors) being associated as risk or protective factors for the clinical and microbiological outcomes of CAZ-AVI-based therapy. Notably, sex (female %), ICU admission, and fever were negatively associated with patents’ mortality serving as independent negative factors for increasing patients’ survival with respective 0.071, 0.141, and 0.069 odds ratios (ORs). Multivariate logistic regression analysis further confirmed that fever experiences in patients, CRP biolevels, and inotropes were the significant independent negative factors (risk factor) that could impact the CAZ-AVI clinical efficiency at an adjusted odds ratios of 0.004, 0.987, and 0.051, respectively. Only ICU admission was positively associated with clinical efficiency serving as positive predictors (OR= 21.183) . Regarding microbial eradication, duration of CAZ-AVI therapy positively influenced the clearance of the CRKP OXA-48-like strain (OR= 1.446). On the contrarily, both WBC counts, and fever experiences were depicted as significant independent covariant/factor that negatively impact the microbial eradication (ORs= 0.747 and 0.013, respectively). All these findings were highlighted within Tables 1-5, thoroughly within the manuscript context as well as summarized in both the abstract and conclusion sections. Kindly refer to Yellow highlights at lines 31-41 and 513-519. Additionally, authors illustrated how observed findings are in good agreement with literature reported studies through data narration within the discussion section. Finally, the authors describe the study’s strengths and limitations with a prospective work for improvement and all were also highlighted. Kindly refer to Yellow highlights at lines 372-374, 383-387, 411-416, and 422-434.

  1. Comments on the Quality of English Language: Scientific spelling errors throughout the manuscript. Easy to read, otherwise. Only minor edits needed.

Response 9: Authors thank the reviewer for valuable comment and great expectation within the presented work. Authors have thoroughly revised the manuscript making it highly appealing and interesting to readers. All typos have been corrected.

Reviewer 3 Report

Comments and Suggestions for Authors

Maamon et al reported a series of 114 infections with Oxa48 Klebsiella treated with either a CZA-containing regimen or an alternative. The report is important given the available articles already published.

A significant amount of clarification, statistics, reporting and writing is required before publication can be considered.

Main points:

Abstract: non-informative study design, nb of patients, nature of the study, setting, statistical analysis, main results.

Grammar and typos: the final text should be thoroughly checked by a native English speaker.

Please also note

Line 63-64: wording

Line 88-89: wording ", received antibiotic 88 regimens for less than 5 days and had bacterial contamination were excluded".

Line 94: "Table 65"???

Methods:

Line 90: exclusion criteria, more precise description needed.

Table 1: co-morbidities should be detailed.

Duration of antibiotic use: unclear to me. Duration of antibiotic therapy for KP oxa48 infections or total duration of antibiotic therapy

Please describe antimicrobials given for CZA and other antibiotic groups? Number of bi antibiotic therapy?

Fever: please give definition?

Definition of clinical remission? Severity of patients not specified? Patients location (ICU? ) bacterial recurrence please define?

Table 2: define intervention?

Statistics:

univariate analysis is not always appropriate.

Multiple linear regression is not an appropriate statistical model to compare 2 groups of patients. It is not an appropriate model to compare survivors and decedents.

Methods: Excluding patients who die before day 5 is a methodological problem called historical bias, because patients may die before day 5 because of the antibiotic therapy given.

The statistics are not appropriate and not well done. It should be done again with the help of a senior statistician.

The discussion should focus on Oxa-48. Too many references dealing with the activity of CZA on KPC, which should be deleted or at least abstracted.

Important references should be cited and commented: Castanheira et al , J Antimicrob Chemother 2021; 76: 3125–3134

 Tamma et al - Clinical Infectious Diseases® 2021;72(7):e169–83;

Paul M et al - Clinical Microbiology and Infection 28 (2022) 521e547

Nagyecar et al – Indian J crit care med 2021 ; 25 :780-84

Another very recent exhaustive text may help to re write the discussion:

Infectious Diseases Society of America 2023 Guidance on the Treatment of Antimicrobial Resistant Gram-Negative Infections.

Tamma PD, Aitken SL, Bonomo RA, Mathers AJ, van Duin D, Clancy CJ.Clin Infect Dis. 2023 Jul 18:ciad428. doi: 10.1093/cid/ciad428. Online ahead of print.PMID: 37463564

Comments on the Quality of English Language

see previous chapter ..;a lot of typos and grammatical mistakes. 

Author Response

Reviewer # 3: Comments and Suggestions

Maamon et al reported a series of 114 infections with Oxa48 Klebsiella treated with either a

CZA-containing regimen or an alternative. The report is important given the available articles

already published. A significant amount of clarification, statistics, reporting and writing is required before publication can be considered. The authors would like to thank the reviewer for the time and effort in revision. Kindly, find the detailed responses and the revised manuscript as per the reviewer’s addressed comments and suggestions.

Main points:

  1. Abstract: non-informative study design, nb of patients, nature of the study, setting, statistical analysis, main results.

Response 1: Authors thank the reviewer for valuable suggestion. The abstract has been modified to include problem and study objectives (Lines 16-18), study design and methodology (Lines 19-28), Results (Lines 28-41), Conclusion and future work (Lines 41-45).

  1. Grammar and typos: the final text should be thoroughly checked by a native English speaker.

Response 2: Authors thank the reviewer for valuable suggestion. Authors have thoroughly revised the manuscript making it highly appealing and interesting to readers. All typos have been corrected.

  1. Please also note Line 63-64: wording

Response 3: Authors thank the reviewer for valuable suggestion. Text has been rephrased and restructured for better understanding and clear message deliver.

  1. Line 88-89: wording ", received antibiotic 88 regimens for less than 5 days and had bacterial contamination were excluded".

Response 4: Authors thank the reviewer for valuable suggestion. Text has been rephrased and restructured for better understanding and clear message deliver. Kindly refer to Lines 467-468

  1. Line 94: "Table 65"???

Response 5: Authors thank the reviewer for valuable comment. Tables and their context within the manuscript have been revised and modified as per the conducted deep analysis including superiority of each treatment arm in terms of patient’s demography, characteristics, site of infection, co-morbidities, details of antibiotic usage, and clinical signs / specific lab values. Comparative analysis was done through applying independent sample t-test or Mann-Whitney U test for normally or non-normally distributed data, respectively. In regard to categorical data, comparisons are expressed in case numbers and percentages and then analyzed using the contingency testing. Multivariable logistic regression analysis was to follow for assessing any potential association of independent variables with CAZ-AVI’s clinical and microbiological efficiencies (clinical success, 30-day all-cause mortality, microbial eradication, bacterial recurrence). Kindly, refer to Tables 1-5.

Methods:

  1. Line 90: exclusion criteria, more precise description needed.

Response 6: Authors thank the reviewer for valuable suggestion. Authors rephrased the patient inclusion and exclusion criteria making it clearer to readers as follows: The study has included; (1) admitted patients aged more than 18 years old in any department of hospital including ICU admitted patients and patients transferred to ICU for treatments, (2) patients with K. pneumonia OXA 48-like gene with any site of infection and being confirmed through drug sensitivity and bacterial cultures, (3) patients who received one or more antibiotics starting from positive culture results and for at least five days. The exclusion criteria were adopted as the following; (1) patients aged younger than 18 years old, (2) patients died prior receiving of the OXA-48 antibiotic therapy or could not be assessed for clinical efficiency (who used antibiotics for less than 5 days), (3) patients lacking the OXA 48-like gene for K. pneumonia. Kindly refer to Section 4.3. of the Methods part.

  1. Table 1: co-morbidities should be detailed.

Response 7: Authors thank the reviewer for valuable suggestion. The authors have categorized the investigated patient of the two-arm treatment groups as per the details of patients’ demography, site of infection, co-morbidities, antibiotic usage details, and clinical sign / special lab results. Co-morbidities were detailed as follows; Respiratory diseases, Cardiovascular disease, Diabetes mellitus, CNS diseases, Cerebrovascular diseases, Kidney diseases, GIT diseases, Septic shock/sepsis, Tumors, and COVID-19 infections. Kindly refer to Tables 1, 3, and 4 and their context within the manuscript.

  1. Duration of antibiotic use: unclear to me. Duration of antibiotic therapy for KP oxa48 infections or total duration of antibiotic therapy

Response 8: Authors thank the reviewer for valuable comment. Duration of the antibiotic therapy is that following the bacterial isolates obtained from the infected patients (first-positive cultures for the OXA-48-like gene in CRKP patients). This was also the defined start point for the 30-day all-cause mortality. Authors made it clearer and represent average antibiotic durations for both treatments arms (CAZ-AVI intervention and standard therapy control groups). Kindly refer to Green highlights at Lines 159-160 and Figure 1A.

  1. Please describe antimicrobials given for CZA and other antibiotic groups? Number of bi antibiotic therapy?

Response 9: Authors thank the reviewer for valuable comment. Standard therapy was monotherapy, combination of two agents, or combination of more than two antibiotics. These included; aminoglycosides for 9 (7.9%), aztreonam for 4 (3.5%), colistin for 61 (53.5%), and tigecycline for 45 (39.5%) of the total patients cases. Details of their frequencies among the investigated patients of both treatment arms (CAZ-AVI intervention and standard therapy control groups) have been thoroughly presented within Table 1, Figure 1B-E, and their manuscript context.

  1. Fever: please give definition?

Response 10: Authors thank the reviewer for valuable comment. Clear definitions and outcomes are now explained within the methodology Section 4.2. Kindly refer to lines 483-494.

  1. Definition of clinical remission? Severity of patients not specified? Patients location (ICU?) bacterial recurrence please define?

Response 11: Authors thank the reviewer for valuable comment. Clear definitions and outcomes are now explained within the methodology Section 4.2. Kindly refer to lines 483-498.

  1. Table 2: define intervention?

Response 12: Authors thank the reviewer for valuable comment. The study investigated the clinical and microbiological outcomes for CAZ-AVI-treated patients as add on to standard therapy (Intervention group) for CRKP OXA-48 infections as compared to CRKP OXA-48 patients received only the standard antibiotic regimen (Comparative group).

Statistics:

  1. univariate analysis is not always appropriate.
  2. Multiple linear regression is not an appropriate statistical model to compare 2 groups of patients. It is not an appropriate model to compare survivors and decedents.

Responses 13 and 14: Authors thank the reviewer for valuable comment. The authors have categorized the investigated patient of the two-arm treatment groups as per the details of patients’ demography, site of infection, co-morbidities, antibiotic usage details, and clinical sign / special lab results. All data have been revised and modified as per the conducted deep analysis including comparing the differential superiority of each treatment arm through applying independent sample t-test or Mann-Whitney U test for normally or non-normally distributed data, respectively. In regard to categorical data, they are expressed in case numbers and percentages and then analyzed using the contingency testing (Chi-square or Fisher’s exact test). Multivariable logistic regression analysis was to follow for assessing any potential association between the independent variables (predictors) with CAZ-AVI’s microbiological and clinical efficiencies. Only variables depicted significance (P < 0.05) throughout the univariate analysis were included within the multivariate logistic regression analysis. Kindly, refer to Tables 1-5.

  1. Methods: Excluding patients who die before day 5 is a methodological problem called historical bias, because patients may die before day 5 because of the antibiotic therapy given.

Response 15: Authors thank the reviewer for valuable comment. Authors made it clearer to reader as being described within the methodology Section 4.3. Among the excluded patients were those who died prior receiving of the OXA-48 antibiotic therapy or could not be assessed for clinical efficiency (who used antibiotics for less than 5 days). As authors further highlights that the duration of antibiotic therapy is that following the bacterial isolates obtained from the infected patients (first-positive cultures for the OXA-48-like gene in CRKP patients)

  1. The statistics are not appropriate and not well done. It should be done again with the help of a senior statistician.

Response 16: Authors thank the reviewer for valuable suggestion. Kindly refer to the above

  1. The discussion should focus on Oxa-48. Too many references dealing with the activity of CZA on KPC, which should be deleted or at least abstracted.

  1. Important references should be cited and commented:

Castanheira et al , J Antimicrob Chemother 2021; 76: 3125–3134 Ref. 52

Tamma et al - Clinical Infectious Diseases® 2021;72(7):e169–83 Ref. 55

Paul M et al - Clinical Microbiology and Infection 28 (2022) 521e547 Ref. 56

Nagvekar et al – Indian J crit care med 2021 ; 25 :780-84 Ref. 67

  1. Another very recent exhaustive text may help to re write the discussion:

Infectious Diseases Society of America 2023 Guidance on the Treatment of Antimicrobial

Resistant Gram-Negative Infections. Tamma PD, Aitken SL, Bonomo RA, Mathers AJ, van

Duin D, Clancy CJ.Clin Infect Dis. 2023 Jul 18:ciad428. doi: 10.1093/cid/ciad428. Online

ahead of print.PMID: 37463564 Ref. 45

Responses 17-19: Authors thank the reviewer for valuable suggestion. The discussion section was enriched by valuable manuscripts suggested by the reviewer. Data regarding current clinical burden and management of OXA-48-like strains are now highlighted within the discussion sections. Data regarding the activity of CAZ-AVI on KPC was abstracted. Comparative narration between the study’s findings in terms of clinical efficiency/success, microbial eradication and mortality were highlighted in relation to data reported by literature studies, including those highlighted by the reviewer. Kindly refer to Green highlights at lines 313-316, 318-324, 335-346, and 390-400.

  1. Comments on the Quality of English Language: see previous chapter ..;a lot of typos and grammatical mistakes.

Response 20: Authors thank the reviewer for valuable comment and great expectation within the presented work. Authors have thoroughly revised the manuscript making it highly appealing and interesting to readers. All typos have been corrected.